# Equilibrium Studies of Iron (III) Complexes with Either Pyrazine, Quinoxaline, or Phenazine and Their Catecholase Activity in Methanol

**DOI:** 10.3390/molecules27103257

**Published:** 2022-05-19

**Authors:** José J. N. Segoviano-Garfias, Gabriela A. Zanor, Fidel Ávila-Ramos, Egla Yareth Bivián-Castro

**Affiliations:** 1División de Ciencias de la Vida (DICIVA), Campus Irapuato-Salamanca, Universidad de Guanajuato, Ex Hacienda El Copal, Carretera Irapuato-Silao Km. 9, Irapuato 36500, Mexico; gzanor@ugto.mx (G.A.Z.); ledifar@ugto.mx (F.Á.-R.); 2Centro Universitario de los Lagos, Universidad de Guadalajara, Enrique Díaz de León 1144, Col. Paseos de la Montaña, Lagos de Moreno 47460, Mexico; egla.bivian@academicos.udg.mx

**Keywords:** iron (III) complexes, catecholase activity, formation constants

## Abstract

Currently, catalysts with oxidative activity are required to create valuable chemical, agrochemical, and pharmaceutical products. The catechol oxidase activity is a model reaction that can reveal new oxidative catalysts. The use of complexes as catalysts using iron (III) and structurally simple ligands such as pyrazine (pz), quinoxaline (qx), and phenazine (fz) has not been fully explored. To characterize the composition of the solution and identify the abundant species which were used to catalyze the catechol oxidation, the distribution diagrams of these species were obtained by an equilibrium study using a modified Job method in the HypSpec software. This allows to obtain also the UV-vis spectra calculated and the formation constants for the mononuclear and binuclear complexes with Fe^3+^ including: [Fe(pz)]^3+^, [Fe_2_(pz)]^6+^, [Fe(qx)]^3+^, [Fe_2_(qx)]^6+^, [Fe(fz)]^3+^, and [Fe_2_(fz)]^6+^. The formation constants obtained were log β_110_ = 3.2 ± 0.1, log β_210_ = 6.9 ± 0.1, log β_110_ = 4.4 ± 0.1, log β_210_ = 8.3 ± 0.1, log β_110_ = 6.4 ± 0.2, and log β_210_ = 9.9 ± 0.2, respectively. The determination of the catechol oxidase activity for these complexes did not follow a traditional Michaelis–Menten behavior.

## 1. Introduction

Identifying catalysts with oxidative activity is an important research area, considering that several oxidation reactions have been used to create valuable chemical, agrochemical, and pharmaceutical products [1]. The catechol oxidase and catecholase are type-III copper metalloenzymes, which catalyze the oxidation of catechol to the corresponding *o*-quinones [1,2]. The understanding and mimicking of the catechol oxidase activity has several purposes: For example, it helps to facilitate the treatment of wastewater contaminated with phenolic compounds [3], and it contribute to the degradation of persistent organic pollutants [4]. In addition, catechol oxidase uses molecular oxygen as a co-substrate, promoting several environmental and economic advantages [1]. Inspired by catechol oxidase, numerous complexes with mono or binuclear coordination sites have been reported, using a wide range of transition metals, such as: Cu, Mn, Co, Ni and Zn. In addition, their activity in the catechol–oxidation reaction and the Michaelis–Menten kinetic parameters of these catalytic systems were previously reported [1,2]. However, iron is the most abundant transition metal found in the Earth’s crust and has multiple functions in several bioinorganic systems [5]. Some biomolecules that contain iron cofactors include hemoglobin and hemerythrin (transport of dioxygen), transferrin (transport of Fe), ferritin (store of Fe), rubredoxin and ferredoxin (catalyze nitrates to ammonia), hydroxylases (hydroxylation), peroxidases (utilization of H_2_O_2_), acid phosphatases (hydrolysis of phosphates groups), superoxide dismutase (dismutation of dioxygen), FUR and FNR (gene regulation), and calcineurin (phosphatase) [6,7]. The catechol oxidase activities of iron complexes have not been extensively measured. Recently, a few iron complexes were obtained and tested as catalysts for catechol oxidation [2,8].

Our objective is to contribute to the understanding regarding the catalytic effects of the mononuclear and binuclear sites of iron (III) in a coordination compound using simple ligands to obtain the kinetic parameters of the oxidation reaction of catechol and correlate with previous research [1]. To achieve this, we studied the complexes created between iron (III) and each of the following ligands: pyrazine (pz), quinoxaline (qx), and phenazine (fz). Pyrazine is a natural compound used to flavor some foods [9,10]; quinoxaline has biological characteristics used in the medical field [11]; and phenazine has been used for research purposes in material sciences and biological applications [12,13]. In this equilibrium study, we reported on the formation constants and the distribution diagram species in mononuclear and binuclear complexes of Fe^3+^ with the individual ligands pyrazine (pz), quinoxaline (qx), and phenazine (fz). These parameters were obtained to determinate which catalytic species were present in a solution under conditions for catechol oxidation. To date, no research has been published identifying the formation constants of iron (III) complexes, containing either pyrazine, quinoxaline, or phenazine, in methanol or any other solvent, or its catalytic activity in catechol oxidation.

## 2. Results and Discussion

### 2.1. Materials and Methods

In the experiments presented in this work, methanol was used in the equilibrium study as well as in the catalytic reactions. We expected similar solvation shells based on the effects of methanol in Fe^3+^ complexes [14] and donor numbers of methanol (19) and water (18) [15]. Methanol also has been shown to solubilize a large amount of dioxygen [16]. In order to use of a wide concentration range of ligands (pyrazine, quinoxaline, and phenazine) and prevent the early precipitation of iron (III) complexes, ionic strength was not employed; therefore, the formation constants in this study should not be considered stability constants and should be used to compare systems measured under similar conditions.

### 2.2. Formation Constants of the Iron (III) Complexes with -Pyrazine, -Quinoxaline, and -Phenazine

To our knowledge, there is no report of the formation constants of iron (III) complexes with -pyrazine, -quinoxaline, and -phenazine, individually, whether in methanol or any other solvent. The electronic spectra of methanol solutions of iron (III)-pyrazine, iron (III)-quinoxaline, and iron (III)-phenazine complexes, are shown in Appendix A, respectively. In all solution systems, several peaks appeared at low Fe^3+^ concentrations; as the concentration of Fe^3+^ increased, a hyperchromic effect was observed. For the iron (III)-pyrazine complex, the peaks appeared at 261 and 353 nm; for the iron (III)-quinoxaline complex, they appeared at 233 and 315 nm; and finally, for the iron(III)-phenazine complex, the peaks appeared at 248 and 362 nm. The calculation of the formation constants corresponded to the equilibrium between each ligand, individually, (i.e., pyrazine, quinoxaline, and phenazine) and Fe^3+^. The formation constant log β_jkl_ was obtained by processing the experimental spectra in HypSpec software [17], which allows to obtain a calculation of the molar absorbance by a linear least-square process with a cycle of non-linear refinement of the equilibrium constants [18,19]. In this process is correlated the spectra of the two experiments with each at distinctive ligand concentration and using two different ranges of Fe^3+^ concentrations. A proposal of an equilibrium model for the generation of colored species in solution and an initial value of its formation constant was initially suggested. For all systems, only three colored species, including Fe^3+^, were found when considering that these ligands could behave binuclearly. The formation constants were achieved using the following model, where L is pz (pyrazine), qx (quinoxaline), or fz (phenazine):Fe^3+^ + L ⇌ [Fe(L)]^3+^ log β_110_(1)
2Fe^3+^ + L ⇌ [Fe_2_(L)]^6+^ log β_210_(2)

A summary of the experimental parameters in this study and the logarithmic values are reported in Table 1. We have observed that the mononuclear and binuclear complexes [Fe(fz)]^3+^ and [Fe_2_(fz)]^6+^, respectively, have higher formation constant values than [Fe(pz)]^3+^, [Fe_2_(pz)]^6+^, [Fe(qx)]^3+^, and [Fe_2_(qx)]^6+^. This behavior was likely related to the increase in the aromatic character of the ligands, which could promote its interaction with iron (III) ions.

An analogous behavior was observed in mononuclear manganese (II) complexes with these ligands [20]. Nevertheless, the binuclear complexes with manganese (II) were not found. This was likely related to the difference between the ionic radii of the manganese (II) (approximately 0.97 Å) [21] and iron (III) (between 0.55 and 0.65 Å) [22]. However, iron (III) is a metal ion with hard acid character [23] that can have a strong affinity for a hard base ligand. By comparing the formation constants of the different ligands used in this study, phenazine had a harder base character than pyrazine and quinoxaline, possibly due to the inductive effect of this ligand [24].

The calculated electronic spectra of iron (III)-pyrazine, iron (III)-quinoxaline, and iron (III)-phenazine in methanol are presented in Figure 1, Figure 2 and Figure 3, respectively. The calculated electronic spectra of the [Fe (pz)]^3+^ showed absorption peaks at 261 nm with ε = 9736 L mol^−1^ cm^−1^ and at 352 nm with ε = 2770 L mol^−1^ cm^−1^. The [Fe_2_ (pz)]^6+^ showed absorption peaks at 261 nm with ε = 14,797 L mol^−1^ cm^−1^ and 356 nm with ε = 5983 L mol^−1^ cm^−1^. For [Fe (qx)]^3+^, the absorption peaks were observed at 233 nm with ε = 38,251 L mol^−1^ cm^−1^ and at 315 nm with ε = 9819 L mol^−1^ cm^−1^. The [Fe_2_ (qx)]^6+^ showed absorption peaks at 233 nm with ε = 34,175 L mol^−1^ cm^−1^, 256 nm with ε = 13,637 L mol^−1^ cm^−1^, 315 nm with ε = 10,931 L mol^−1^ cm^−1^, and 346 nm with ε = 8820 L mol^−1^ cm^−1^. For [Fe (fz)]^3+^, the absorption peaks were observed at 248 nm with ε = 105,850 L mol^−1^ cm^−1^ and 362 nm with ε = 18,147 L mol^−1^ cm^−1^. The [Fe_2_ (fz)]^6+^ had absorption peaks at 244 nm with ε = 140,570 L mol^−1^ cm^−1^, at 250 nm with ε = 121,270 L mol^−1^ cm^−1^, and at 360 nm with ε = 28,460 L mol^−1^ cm^−1^. Finally, the calculation of the molar absorbance of pyrazine, quinoxaline, and phenazine in methanol had been previously reported [20].

There is a remarkable difference in the molar absorption values of the complexes, in which the iron (III)-phenazine complex generates higher molar absorption values than the pyrazine or the quinoxaline complexes, which was likely related to the increasing aromatic character of the ligand. For the iron (III) complexes, the peaks at 261 nm for [Fe (pz)]^3+^, 261 nm for [Fe_2_ (pz)]^6+^, 233 nm for [Fe (qx)]^3+^, 233 and 256 nm for [Fe_2_ (qx)]^6+^, 248 nm for [Fe (fz)]^3+^, and 244 and 250 nm for [Fe_2_ (fz)]^6+^ were assigned to π→π* [25,26]. The absorption peaks at 352 nm for [Fe (pz)]^3+^, 356 nm for [Fe_2_ (pz)]^6+^, 315 nm for [Fe (qx)]^3+^, 315 and 346 nm for [Fe_2_ (qx)]^6+^, 362 nm for [Fe (fz)]^3+^, and 360 nm for [Fe_2_ (fz)]^6+^ were assigned to n→π* [25,26], (a resume of their signals and their assignment is in Table 2). The molar absorbance of binuclear complexes can be usually manifested as an increase of the molar absorption coefficient, the maximum wavelength of the complexes is maintained at about the same value, and an analogous behavior has been reported before [27,28].

### 2.3. Distribution Curves of the Iron (III)-Complexes with -Pyrazine, -Quinoxaline, and -Phenazine

The speciation diagrams of the iron (III)-pyrazine, iron (III)-quinoxaline, and iron (III)-phenazine systems are shown in Appendix A, respectively. A solution with equimolar concentrations of iron (III) and pyrazine (at 0.000050 and 0.000075 M, for low and high concentration experiments, respectively) roughly yielded 6.2% of [Fe (pz)]^3+^, approximately 92.16% of free pyrazine, and 1.6% of [Fe_2_ (pz)]^6+^, yet two molar equivalents of iron (III) per pyrazine generated approximately 11.1% of the of [Fe (pz)]^3+^, 5.8% of [Fe_2_ (pz)]^6+^, and 83.12% of free pyrazine. Three molar equivalents of iron (III) per pyrazine generated approximately 14.5% of the of [Fe (pz)]^3+^, 11.0% of [Fe_2_ (pz)]^6+^, and 75% of free pyrazine. According to Appendix A, the solutions with equimolar concentrations of iron (III) and quinoxaline (at 0.00007968 and 0.00015936 M, for low and high concentration experiments, respectively) generated approximately 26% of [Fe (qx)]^3+^, 4% of [Fe_2_ (qx)]^6+^ and 70% quinoxaline. Two molar equivalents of iron (III) per quinoxaline generated approximately 38% of [Fe (qx)]^3+^, 12% of [Fe_2_ (qx)]^6+^ and 50% of free quinoxaline. In addition, the three molar equivalents of iron (III) per quinoxaline generated approximately 42% of [Fe (qx)]^3+^, 22% of [Fe_2_ (qx)]^6+^, and 36% of free quinoxaline. Finally, in Appendix A, the solutions with an equimolar concentration of iron (III) and phenazine (at 0.0000175 and 0.0000351 M, for low and high concentration experiments, respectively) generated approximately 87% of [Fe (fz)]^3+^, 1% of [Fe_2_ (fz)]^6+^, and 12% of free phenazine. The two molar equivalents of iron (III) per phenazine generated approximately 92% of [Fe (fz)]^3+^, 7% of [Fe_2_ (fz)]^6+^, and 2% of free phenazine. Finally, the three molar equivalents of iron (III) per phenazine generated approximately 86% of [Fe (fz)]^3+^, 13% of [Fe_2_ (fz)]^6+^, and 1% of free phenazine.

### 2.4. Far- and Mid-Infrared Spectrum of the Complexes [Fe (pz)]^3+^, [Fe (qx)]^3+^, [Fe (fz)]^3+^, [Fe_2_ (pz)]^6+^, [Fe_2_ (qx)]^6+^, and [Fe_2_ (fz)]^6+^

Appendix A present the far- and mid-infrared spectra for the different complexes obtained in this study. The complexes [Fe (pz)]^3+^, [Fe (qx)]^3+^, [Fe (fz)]^3+^, [Fe_2_ (pz)]^6+^, [Fe_2_ (qx)]^6+,^ and [Fe_2_ (fz)]^6+^ showed bands in the region of 465–350 cm^−1^, which corresponded to ν (M-N) vibrations [29,30,31,32]. In addition, all the complexes in this study presented bands of approximately 250 to 200 cm^−1^, which were assignable to ν (Fe-O) of nitrate complexes [30,31,32]. The complexes with pyrazine, quinoxaline, and phenazine presented signals between 1600 and 1200 cm^−1^ that were associated to ring vibrations of the ligands [33], also nitrate ion signals appeared between 1200 to 1500 cm^−1^ [34]. In addition, signals in the vicinity of 3000 cm^−1^ were assigned to C-H stretching [35,36].

### 2.5. Catecholase Activity of the Complexes with Iron (III) and Pyrazine, Quinoxaline, and Phenazine

The catalytic oxidation of 3,5-di-tert-butyl catechol (DTBC) using atmospheric dioxygen has been widely studied as a model reaction in catecholase (or catechol-oxidase) activity. This substrate can be oxidized to 3,5-di-tert-butyl-*o*-benzoquinone (DTBQ) while the iron (III) catalyst is reduced to a previous oxidation state before undergoing additional reoxidation [2]. This reaction recorded the increase of the absorption of DTBQ at 400 nm during a period of time (Appendix A). The initial rate method was used to study the kinetic profile and analyze its behavior according to the Michaelis–Menten model. Figure 4 presents the initial rate results of the oxidation reaction of the DTBC catalyzed by the complexes [Fe (pz)]^3+^, [Fe (qx)]^3+^, and [Fe (fz)]^3+^; when the substrate concentration increased, the rate decreased with oscillations. Figure 5 presents the initial rate results of the oxidation of DTBC catalyzed by the complexes [Fe_2_ (pz)]^6+^, [Fe_2_ (qx)]^6+^, and [Fe_2_ (fz)]^6+^, in which we found that as the substrate concentration increased, its rates fluctuated. The Michaelis–Menten model has been used to predict a catalytic reaction rate by the possible formation of a catalyst–substrate complex, in which an increase of substrate concentration increases also the reaction rate. Also, increasing unbinding of catalyst–substrate complexes will decrease the reaction rate. Under some experimental conditions, an increase of substrate concentration might promote an unbinding effect on catalytic reactions and reduce the rate of product formation. Nevertheless, the development of an accurate mathematical model predicting an inhibitory unbinding has not been fully understood due to the lack of experimental data [37]. The results presented here indicated that the reactions infringed on the traditional Michaelis–Menten behavior. The generation of further experimental data for diverse catalytic systems could promote the development of complementary mathematical models including the role of unbinding substrates [37]. Considering that pyrazine, quinoxaline, and phenazine could behave as π-electron attracting ligands [38], it must be evaluated the possible generation of a stable adduct between the mononuclear and binuclear catalytic complexes and DTBC and DTBQ. In order to evaluate this theory, the possible adduct must be isolated and a crystallographic study should be done. Also, these kinetic reactions should be repeated in the presence of a radical scavenger and a radical promoter to get more insight into the nature of the intermediary.

## 3. Experimental

### 3.1. Materials and Methods

For the preparation of the different solutions, methanol HPLC grade (Fermont, Mexico) was used as solvent, Fe (NO_3_)_3_·9H_2_O (Sigma-Aldrich, St. Louis, MI, USA), pyrazine (Sigma-Aldrich, USA), quinoxaline (Sigma-Aldrich, St. Louis, MI, USA), phenazine (Sigma-Aldrich, St. Louis, MI, USA), and 3,5-di-tert-butylcatechol (DTBC) (Sigma-Aldrich, St. Louis, MI, USA) were analytical grade and used without further purification. The spectral measurements were performed at 298 K in a quartz cell with a 1 cm path length and 3 mL volume. We used a Shimadzu UV-vis-1800 spectroscopy system equipped with a Thermo Scientific TPS-1500W thermostat system. The spectrophotometric data were fitted in HypSpec [18,19], and the distribution diagrams of species were calculated in the Hyperquad Simulation and Speciation software (HySS2009, Leeds, UK) [39].

In a typical spectral measurement of the iron (III)-ligand complexes, stock solutions of the ligand (either pyrazine, quinoxaline, or phenazine) and iron (III) were prepared and diluted to obtain a solution behaving according to the Beer–Lambert law. The final concentration of the ligand was constant, and the concentration of Fe (III) ion was varied within a range. This process was repeated twice, and each experiment was conducted at a different concentration of the ligand and at two different concentration ranges of Fe (III) ions. In this manuscript, we reported only the experiments with concentration ranges that allowed us to obtain spectral measurements of the interactions between the mononuclear and binuclear complexes of Fe (III)-ligands without precipitation. For the determination of the formation constants in all the experiments, the spectral region analyzed was from 220 to 500 nm.

### 3.2. Equilibrium Studies of Iron (III) with Pyrazine, Quinoxaline, or Phenazine

The experiments were conducted using stock solutions at different concentrations of the iron (III) and ligands (either pyrazine, quinoxaline, or phenazine). For each system, the ligand concentration was constant, and the concentration of the Fe (III) ion was varied; the UV-vis spectrum was recorded for the different solutions. For all solution systems, the final ligand concentration remained constant in the two experiments at 24.58 and 36.87 µM. In each experiment the iron (III) concentration was varied from 2.57 to 74.53 µM and from 3.96 to 114.84 µM, respectively.

### 3.3. Synthesis of the Complexes: [Fe (pz)]^3+^, [Fe (qx)]^3+^, [Fe (fz)^3+^, [Fe_2_ (pz)]^6+^, [Fe_2_ (qx)]^6+^ and [Fe_2_ (fz)]^6+^ and Its Far- and Mid-Infrared Spectrum

The Fe^3+^ complexes were prepared in methanol HPLC (100 mL) by combining the iron (III) nitrate nonahydrate and the ligand used (either pyrazine, quinoxaline, or phenazine). According to the distribution diagrams, to prepare the [Fe (ligand)]^3+^, a two-molar solution of iron (III) nitrate nonahydrate (5.0 mM) and a molar solution of the ligand (2.5 mM) were mixed in methanol. In addition, to prepare the [Fe_2_ (ligand)]^6+^, a three-molar solution of iron (III) nitrate nonahydrate (7.5 mM) was mixed with a molar solution of the ligand (2.5 mM). The solutions remained at 4 °C until precipitation, and later the solutions were filtered to collect the product. If solid samples were redissolved and its UV-vis spectrum obtained, the signals were comparable to the UV-vis spectrum calculated for the complexes. The far- and mid-infrared spectra for the different iron (III) complexes were obtained using an HATR system in a Perkin–Elmer Frontier FTIR/FIR spectrometer in the range of 700 to 50 and 1400 to 400 cm^−1^, respectively.

### 3.4. Catecholase Activity of the Complexes with Iron (III) and Pyrazine, Quinoxaline, and Phenazine

The catecholase activity of the complexes was analyzed using the initial rates method and monitoring spectrophotometrically for the oxidation of 3,5-di-tert-butyl catechol (DTBC) to 3,5-di-tert-butyl-*o*-benzoquinone (DTBQ) at 400 nm (ε = 1900 mol^−1^ L cm^−1^) [40] in methanol solution at 298 K. Stock solutions of Fe (NO_3_)_3_·9H_2_O, pyrazine, quinoxaline, and phenazine were prepared separately and diluted in order to obtain a final concentration (0.000015 M of the ligand and 0.000030 or 0.000045 M for Fe^3+^ ion to generate the mononuclear and binuclear complexes, respectively); 3,5-di-tert-butyl catechol was prepared at 0.045 M and diluted to a range of 0.029–0.040 M.

## 4. Conclusions

The ligands pyrazine, quinoxaline, and phenazine, allows to bind two Fe (III) ions. As the aromatic character in the ligand increased, the affinity for the iron (III) site also increases. The structural simplicity of these ligands and their use in iron (III) complexes still requires an understanding of their binding sites, spectral properties, and catalytic activities. Nevertheless, by comparing the equilibrium, spectral and kinetic parameters of the iron (III) complexes, allows to increase the understanding of the catalytic complex and its affinity for the substrate to make a better proposal of the ligands to be used as iron (III) catalysts, evolving the catecholase model complex. The affinity of these ligands, in particular phenazine, for iron (III) ion suggests a synergism between the ligand and the iron (III) ion, promoting a strong spectral effect. In addition, there may be an affinity of the complex for the substrate that might decrease its reaction rate in catecholase activity. The binuclear complexes with an aromatic ligand might have a greater affinity for the substrate, which may be related to the aromatic character of the ligand. In order to confirm this theory, further experimental research should be conducted: characterization of iron (III) complexes using X-ray diffraction, also an equilibrium study of iron (III) complexes with the DTBC which will allow to evaluate the possibility of the formation of DTBC-dimeric intermediates and its possible hyperchromism, should be done.

## Figures and Tables

**Figure 1 molecules-27-03257-f001:**
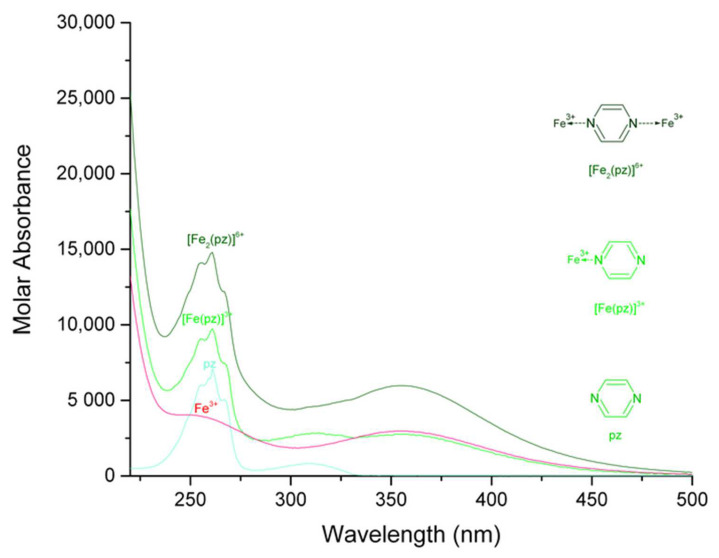
Calculated electronic spectrum of the solution system iron (III)-pyrazine in methanol: Fe^3+^, [Fe (pz)]^3+^, [Fe_2_ (pz)]^6+^, and pz.

**Figure 2 molecules-27-03257-f002:**
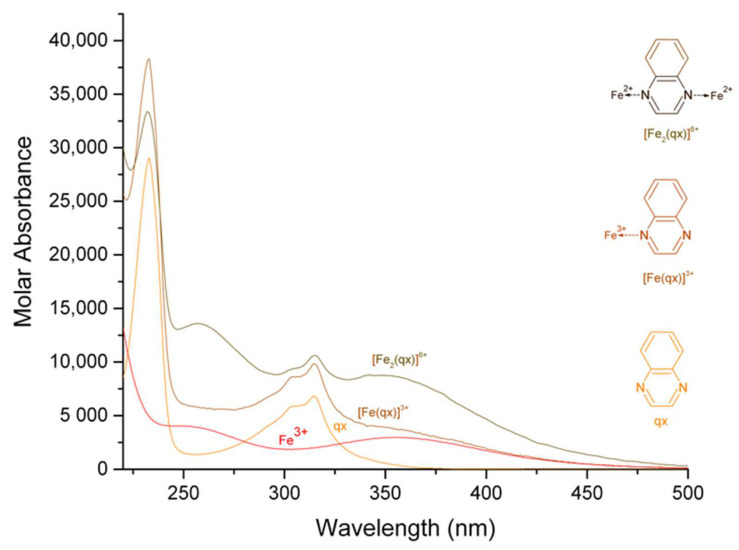
Calculated electronic spectrum of the solution system iron (III)-quinoxaline in methanol: Fe^3+^, [Fe (qx)]^3+^, [Fe_2_ (qx)]^6+^, and qx.

**Figure 3 molecules-27-03257-f003:**
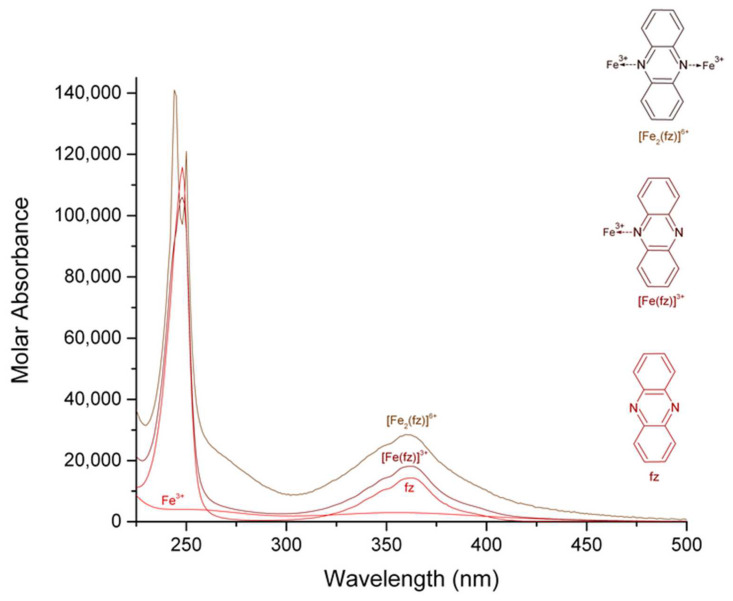
Calculated electronic spectrum of the solution system iron (III)-phenazine in methanol: Fe^3+^, [Fe (fz)]^3+^, [Fe_2_ (fz)]^6+^, and fz.

**Figure 4 molecules-27-03257-f004:**
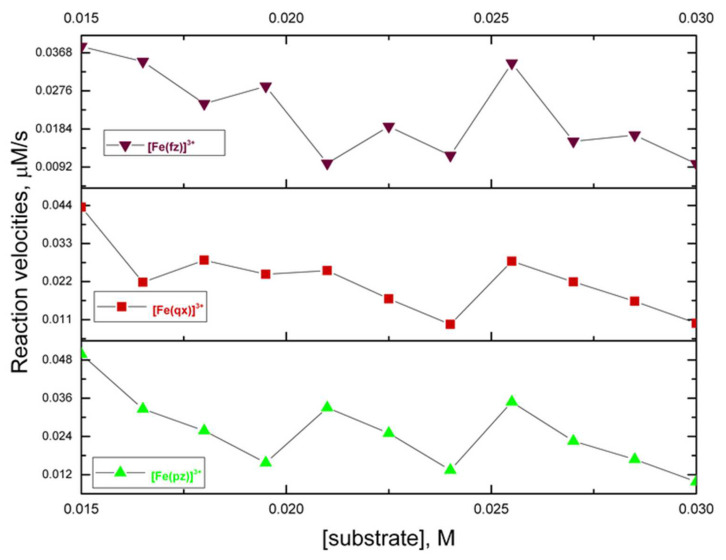
Initial rate results on the oxidation reaction of DTBC to DTBQ and catalyzed by [Fe (pz)]^3+^, [Fe (qx)]^3+^, and [Fe (fz)]^3+^.

**Figure 5 molecules-27-03257-f005:**
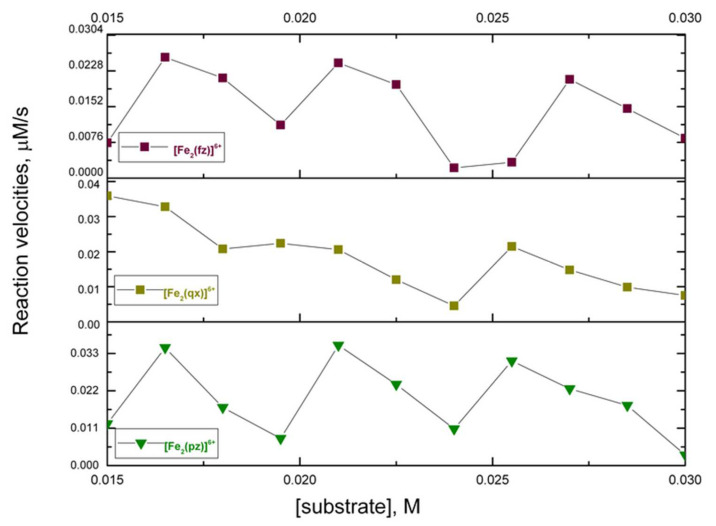
Initial rate results on the oxidation reaction of DTBC to DTBQ and catalyzed by [Fe_2_ (pz)]^6+^, [Fe_2_ (qx)]^6+^, and [Fe_2_ (fz)]^6+^.

**Table 1 molecules-27-03257-t001:** Summary of experimental parameters for the systems Fe^3+^ with pyrazine (pz), quinoxaline (qx) and phenazine (fz) in methanol.

Solution Composition		[T_L_] Constant at 50 and 75 µmol L^−1^ [T_M_] Range from 7.5 to 148.8 and 11 to 212 µmol L^−1^	
		Ionic strength, electrolyte	Not used
		pH range	Not used
Experimental method		Spectrophotometric titration	
Temperature		298 K	
Total number of data points		Fe complexation: 27 solution spectra	
Method of calculation		HypSpec	
Species	Equilibrium	Log β	σ
[Fe (pz)]^3+^ [Fe_2_ (pz)]^6+^	Fe^3+^ + pz ⇌ [Fe (pz)]^3+^ 2Fe^3+^ + pz ⇌ [Fe_2_ (pz)]^6+^	log β_110_ = 3.2 ± 0.1 log β_210_ = 6.9 ± 0.1	0.0019
Solution composition		[T_L_] constant at 24.85 and 36.87 µmol L^−1^ [T_M_] range from 2.6 to 74.5 and 4.0 to 114.8 µmol L^−1^	
		Ionic strength, electrolyte	Not used
		pH range	Not used
Experimental method		Spectrophotometric titration	
Temperature		298 K	
Total number of data points		Fe complexation: 27 solution spectra	
Method of calculation		HypSpec	
Species	Equilibrium	Log β	σ
[Fe (qx)]^3+^ [Fe_2_ (qx)]^3+^	Fe^3+^ + qx ⇌ [Fe (qx)]^3+^ 2Fe^3+^ + qx ⇌ [Fe_2_ (qx)]^6+^	log β_110_ = 4.4 ± 0.1 log β_210_ = 8.3 ± 0.1	0.0019
Solution composition		[T_L_] constant at 24.85 and 36.87 µmol L^−1^ [T_M_] range from 2.6 to 74.5 and 4.0 to 114.8 µmol L^−1^	
		Ionic strength, electrolyte	Not used
		pH range	Not used
Experimental method		Spectrophotometric titration	
Temperature		298 K	
Total number of data points		Fe complexation: 27 solution spectra	
Method of calculation		HypSpec	
Species	Equilibrium	Log β	σ
[Fe (fz)]^3+^ [Fe_2_ (fz)]^3+^	Fe^3+^ + fz ⇌ [Fe (fz)]^3+^ 2Fe^3+^ + fz ⇌ [Fe_2_ (fz)]^6+^	log β_110_ = 6.4 ± 0.2 log β_210_ = 9.9 ± 0.2	0.0017

**Table 2 molecules-27-03257-t002:** UV-vis spectral data for the complexes [Fe (pz)]^3+^, [Fe (qx)]^3+^, [Fe (fz)]^3+^, [Fe_2_ (pz)]^6+^, [Fe_2_ (qx)]^6+^, and [Fe_2_fz)]^6+^.

Complex	Maximum Wavelength (nm); Molar Absorbance (L mol^−1^ cm^−1^)	Assignment, Reference	Maximum Wavelength (nm); Molar Absorbance (L mol^−1^ cm^−1^)	Assignment, Reference
[Fe (pz)]^3+^	261; 9736	π→π* [25,26]	352; 2770	n→π* [25,26]
[Fe (qx)]^3+^	233; 38,251		315; 9819	
[Fe (fz)]^3+^	248; 105,850		362; 18,147	
[Fe_2_ (pz)]^6+^	261; 14,797		356; 5983	
[Fe_2_ (qx)]^6+^	233, 256; 34,175, 13,637		315, 346; 10,931, 8820	
[Fe_2_ (fz)]^6+^	244, 250; 140,570, 121,270		360; 28,460	

## Data Availability

The data presented in this study are available in Appendix A.

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
