# Peer review of "Equilibrium Studies of Iron (III) Complexes with Either Pyrazine, Quinoxaline, or Phenazine and Their Catecholase Activity in Methanol"

_molecules, 2022, doi:10.3390/molecules27103257_

Round 1
Reviewer 1 Report
A full paper “Equilibrium Studies of Iron(III) Complexes with Pyrazine, Quinoxaline or Phenazine and its Catecholase Activity in Methanol” by José J. N. Segoviano-Garfias and the colleagues is devoted to the study of complexation reactions of Fe(NO3)3·9H2O and several linking ligands (pyrazine, quinoxaline or phenazine). Resulted compounds were evaluated as catalysts in 3,5-di-tert-butyl catechol oxidation. In principle, these results may be of interest for Molecules readership. I may recommend it for the publication on terms of major revision.
My remarks are as follows:
1. Formation of complexes of iron nitrate and ligands is not very proved. Compounds from the article may be viewed as simple mixture of two substances. Complex formation now is shown just as precipitate observation. Could Mössbauer spectroscopy (XANES, X-ray diffraction) be used for the characterization of complexes? It will be great to see spectrum of reaction mixture and sum of spectra of ligands and iron (hyperchromism).
- FTIR instrument and the style of IR spectra registration are not mentioned in experimental part. Assignment of peaks includes NH signals but ligands do not contain NH fragments. In turn, NO3 groups are not discussed (resulted complexes should have a lot of them).
- Please add a spectrum with DTBQ absorption. Regarding the discussion concerning nature of catalytic species – note that even polynuclear iron complexes during oxidations tend to form dimeric intermediates https://doi.org/10.1002/anie.201607189
Author Response
Response to Reviewer 1 Comments
Dear Reviewer:
Thank you so much for reviewing the manuscript: Equilibrium Studies of Iron(III) Complexes with either Pyrazine, Quinoxaline, or Phenazine and their Catecholase Activity in Methanol.
Point 1: Formation of complexes of iron nitrate and ligands is not very proved. Compounds from the article may be viewed as simple mixture of two substances. Complex formation now is shown just as precipitate observation. Could Mössbauer spectroscopy (XANES, X-ray diffraction) be used for the characterization of complexes? It will be great to see spectrum of reaction mixture and sum of spectra of ligands and iron (hyperchromism).
Response 1: Considering the time response of the publisher to attend the reviewer comments(10 days), it is unlikely that XANES or x-ray diffraction can be performed. However, the use of x-ray diffraction and the analyze of an hyperchromism effect, to complete the characterization of the complexes, was included in future perspectives. Also ν(Fe-N) bond vibrations represents a strong evidencie of the complex formation.
Point 2:FTIR instrument and the style of IR spectra registration are not mentioned in experimental part. Assignment of peaks includes NH signals but ligands do not contain NH fragments. In turn, NO3 groups are not discussed (resulted complexes should have a lot of them).
Response 2:An explanation of the HATR system used to obtained the FIR and MIR spectra was added to the experimental part. Also, the asssignment of NH peaks was removed, instead CH peaks was added. Also A brief discussion of nitrate ion in the mid infrared was added.
Point 3:Please add a spectrum with DTBQ absorption. Regarding the discussion concerning nature of catalytic species – note that even polynuclear iron complexes during oxidations tend to form dimeric intermediates https://doi.org/10.1002/anie.201607189
Response 3:Figure S5, was added has added an example of the growing curves for the catalytic oxidation of the complexes for the 3,5-di-tert-butyl catechol to 3,5-di-tert-butyl-o-benzoquinone reaction at 400 nm. Also as a pfuture perspective was added the exploration of iron(III) dimeric intermediates. Considering the subtrate:catalyst ratio, a disturbance of 3,5-di-tert-butyl-o-benzoquinone spectra by the reaction intermediates, is unlikely to be observed.
Reviewer 2 Report
The article investigates formation constants for the formation of Fe(III) mononuclear and dinuclear complexes with pyrazine (pz), quinoxaline (qx) and phenazine (fz) ligands, using UV spectrophotometric titration. The UV spectra of the complexes [Fe(III)L] and [Fe(III)2L] were calculated, where L = pz, qx and fz. To a solution of the Fe(III) salt, varying amounts of the ligands were added and UV spectra recorded. The formation constants were calculated using the software “HypSpec”. The formation constants increased in the order pz < qx < fz. Far- and mid- infrared spectra were also recorded. The complexes were also employed as catalyst in the oxidation of catechol. Using a Michaelis-Menten kinetic model to investigate catecholase activity. Initial rates showed an oscillating behavior, with no clear trend obvious.
In general, the chemistry described in the manuscript is of interest, as iron-catalyzed oxidation reactions are an important alternative to other oxidation methods. The title reaction (i.e. the oxidation of catechol) has previously been demonstrated to be catalyzed by iron. The authors should acknowledge previous work more comprehensively, e.g. Turkish Journal of Chemistry 2016, 40, 588-601; ChemistrySelect 2016, 1, 1910-1916; European Journal of Inorganic Chemistry 2015, 3478-3484.
Still, the manuscript provides important data points, that may be of interest to a specialized audience.
However, I encourage the authors to consider the following points before the manuscript can be published.
As far as I understand, the authors calculated the UV spectra of the complexes [Fe(III)L] and [Fe(III)2L], and used the calculated spectra with the software “HypSpec”. First of all, to me the spectra of [Fe(III)L] and [Fe(III)2L] look very similar, and I cannot see how the formation of [Fe(III)L] can be distinguished from [Fe(III)2L]. The data in Figures S1 to 3 look to me more like the formation of one species, and its concentration increases with increasing Fe concentration. Furthermore, not knowing the software “HypSpec”, I do not know how the spectra actually were converted to formation constants. As such, I cannot really understand how the core data of the manuscript were obtained. The authors should at least describe the method employed by the software. Still, I am not certain whether the calculated spectra allow for the distinction between the mono- and dinuclear species. So do not the near-IR data. Some clarification of these points would be very important.
The authors determined that the catechol oxidation does not follow Michaelis-Menten kinetic. The me, the data in Figure 5 look “random”. The authors assume an iron-substrate complex as the rate-determining intermediate. However, is it possible that the reaction follows a radical mechanism? That may explain the outcome in Figure 5. If a radical reaction is in place, there would be no iron-substrate complex in the catalytic cycle, and the discussion in the conclusion section would be obsolete. Any discussion of an iron-substrate complex is pure speculation. Does the reaction proceed in the presence of radical scavengers?
The manuscript could be presented a little better. A lot of information is shifted to the supporting information. At least Table S1, which contains crucial data, should be moved to the main manuscript, and the findings should be discussed. Some of the UV absorptions and the distributions discussed in the text could be tabulated to make it easier to understand for the audience. In the abstract, the method to obtain formation constants should be mentioned.
Technical points. o-quinones, “o” italicized? “[Fe2(qx)]6+”, 2 subscript.
Some of the references use the abbreviation “et al.” (e.g. #20, 26), which is not good practice. All authors should be listed in the references.
The language might need here and there some improvement, e.g. “Inspired by catechol oxidase, have been reported numerous complexes, using a wide range of transition metals, such as Cu, Mn, Co, Ni and Zn, in a diversity of mononuclear or binuclear coordination sites” better “Inspired by catechol oxidase, numerous complexes with mono or binuclear coordination sites have been reported, using a wide range of transition metals, such as Cu, Mn, Co, Ni and Zn”?
Author Response
Response to Reviewer 2 Comments
Dear Reviewer:
Thank you so much for reviewing the manuscript: Equilibrium Studies of Iron(III) Complexes with either Pyrazine, Quinoxaline, or Phenazine and their Catecholase Activity in Methanol.
Point 1: The article investigates formation constants for the formation of Fe(III) mononuclear and dinuclear complexes with pyrazine (pz), quinoxaline (qx) and phenazine (fz) ligands, using UV spectrophotometric titration. The UV spectra of the complexes [Fe(III)L] and [Fe(III)2L] were calculated, where L = pz, qx and fz. To a solution of the Fe(III) salt, varying amounts of the ligands were added and UV spectra recorded. The formation constants were calculated using the software “HypSpec”. The formation constants increased in the order pz < qx < fz. Far- and mid- infrared spectra were also recorded. The complexes were also employed as catalyst in the oxidation of catechol. Using a Michaelis-Menten kinetic model to investigate catecholase activity. Initial rates showed an oscillating behavior, with no clear trend obvious.
In general, the chemistry described in the manuscript is of interest, as iron-catalyzed oxidation reactions are an important alternative to other oxidation methods. The title reaction (i.e. the oxidation of catechol) has previously been demonstrated to be catalyzed by iron. The authors should acknowledge previous work more comprehensively, e.g. Turkish Journal of Chemistry2016, 40, 588-601; ChemistrySelect 2016, 1, 1910-1916; European Journal of Inorganic Chemistry 2015, 3478-3484.
Still, the manuscript provides important data points, that may be of interest to a specialized audience.
However, I encourage the authors to consider the following points before the manuscript can be published.
As far as I understand, the authors calculated the UV spectra of the complexes [Fe(III)L] and [Fe(III)2L], and used the calculated spectra with the software “HypSpec”. First of all, to me the spectra of [Fe(III)L] and [Fe(III)2L] look very similar, and I cannot see how the formation of [Fe(III)L] can be distinguished from [Fe(III)2L]. The data in Figures S1 to 3 look to me more like the formation of one species, and its concentration increases with increasing Fe concentration. Furthermore, not knowing the software “HypSpec”, I do not know how the spectra actually were converted to formation constants. As such, I cannot really understand how the core data of the manuscript were obtained. The authors should at least describe the method employed by the software. Still, I am not certain whether the calculated spectra allow for the distinction between the mono- and dinuclear species. So do not the near-IR data. Some clarification of these points would be very important.
The authors determined that the catechol oxidation does not follow Michaelis-Menten kinetic. The me, the data in Figure 5 look “random”. The authors assume an iron-substrate complex as the rate-determining intermediate. However, is it possible that the reaction follows a radical mechanism? That may explain the outcome in Figure 5. If a radical reaction is in place, there would be no iron-substrate complex in the catalytic cycle, and the discussion in the conclusion section would be obsolete. Any discussion of an iron-substrate complex is pure speculation. Does the reaction proceed in the presence of radical scavengers?
The manuscript could be presented a little better. A lot of information is shifted to the supporting information. At least Table S1, which contains crucial data, should be moved to the main manuscript, and the findings should be discussed. Some of the UV absorptions and the distributions discussed in the text could be tabulated to make it easier to understand for the audience. In the abstract, the method to obtain formation constants should be mentioned.
Technical points. o-quinones, “o” italicized? “[Fe2(qx)]6+”, 2 subscript.
Some of the references use the abbreviation “et al.” (e.g. #20, 26), which is not good practice. All authors should be listed in the references.
The language might need here and there some improvement, e.g. “Inspired by catechol oxidase, have been reported numerous complexes, using a wide range of transition metals, such as Cu, Mn, Co, Ni and Zn, in a diversity of mononuclear or binuclear coordination sites” better “Inspired by catechol oxidase, numerous complexes with mono or binuclear coordination sites have been reported, using a wide range of transition metals, such as Cu, Mn, Co, Ni and Zn”?
Response 1: The following articles were cited within the manuscript: Turkish Journal of Chemistry2016, 40, 588-601; ChemistrySelect 2016, 1, 1910-1916; European Journal of Inorganic Chemistry 2015, 3478-3484.
At the end of the section 2.2 was added a small paragraph about the absorbance behavior of binuclear complexes, also two specialized references concerning the use of the HypSpec method and its equations for the determination of the equilibrium contants and the calculation of the UV-vis spectrum were added.
Considering the time response of the publisher to attend the reviewer comments(10 days), some future perspectives were added as the use a radical scavenger and a radical promoter in order to compare de reaction rate and get more insight of the nature of the intermediary.
Table S1 were changed to Table 1 and Table 2 were added with a summary of the UV-vis characterization. Also, several changes at the manuscritp were made: in the Abstract were added the method for obtainnig the formation constants, o-for orto was italicized, et al was removed and all authors are mentioned. Also, the paragraph which mentioned: “Inspired by catechol oxidase, have been reported …” was changed as suggested, also english language was revisited.
Round 2
Reviewer 1 Report
Authors improved the manuscript properly. I would recommend it for the publication